# Elderly Gliobastoma Patients: The Impact of Surgery and Adjuvant Treatments on Survival: A Single Institution Experience

**DOI:** 10.3390/brainsci12050632

**Published:** 2022-05-11

**Authors:** Francesco Bruno, Alessia Pellerino, Edoardo Pronello, Rosa Palmiero, Luca Bertero, Cristina Mantovani, Andrea Bianconi, Antonio Melcarne, Diego Garbossa, Roberta Rudà

**Affiliations:** 1Division of Neuro-Oncology, Department of Neuroscience, University and City of Health and Science, 10100 Turin, Italy; alessia.pellerino@unito.it (A.P.); pronello.edoardo@gmail.com (E.P.); rosapalmiero61@gmail.com (R.P.); rudarob@hotmail.com (R.R.); 2Neurology Unit, Department of Translational Medicine, University of Eastern Piedmont, 28100 Novara, Italy; 3Pathology Unit, Department of Medical Sciences, University and City of Health and Science, 10100 Turin, Italy; luca.bertero@unito.it; 4Division of Radiotherapy, Department of Oncology, University and City of Health and Science, 10100 Turin, Italy; cristina.mantovani@gmail.com; 5Division of Neurosurgery, Department of Neuroscience, University and City of Health and Science, 10100 Turin, Italy; andrea.bianconi@unito.it (A.B.); anmelcarne@gmail.com (A.M.); diego.garbossa@unito.it (D.G.); 6Department of Neurology, Castelfranco and Treviso Hospitals, 31100 Treviso, Italy

**Keywords:** glioblastoma, elderly patients, extent of resection, gross-total resection, *MGMT*p methylation, adjuvant treatments, radio-chemotherapy, comorbidity, survival, outcome

## Abstract

*Introduction*. Elderly glioblastoma (GBM) patients often show limited response to treatment and poor outcome. Here, we provide a case series of elderly GBM patients from our Institution, in whom we assessed the clinical characteristics, feasibility of surgical resection, response to adjuvant treatments, and outcome, along with the impact of comorbidities and clinical status on survival. *Patients and Methods*. We included patients ≥ 65-year-old. We collected information about clinical and molecular features, extent of resection, adjuvant treatments, treatment-related complications, and outcome. Results. We included 135 patients. Median age was 71 years. In total, 127 patients (94.0%) had a Karnofsky Performance Status (KPS) ≥70 and 61/135 (45.2%) a Charlson Comorbidity Score (CCI) > 3. *MGMT*p methylation was found in 70/135 (51.9%). Subtotal resections (STRs), gross-total resections (GTRs), and biopsies were 102 (75.6%), 10 (7.4%) and 23 (17.0%), respectively. Median progression-free survival and overall survival (mOS) were 8.0 and 10.5 months for the whole cohort. Notably, GTR and radio-chemotherapy with temozolomide in patients with *MGMT*p methylation were associated with significantly longer mOS (32.8 and 44.8 months, respectively). In a multivariable analysis, risk of death was affected by STR vs. GTR (HR 2.8, *p* = 0.002), *MGMT*p methylation (HR 0.55, *p* = 0.007), and KPS at baseline ≥70 (HR 0.43, *p* = 0.031). Conversely, CCI and post-surgical complications were not significant. *Conclusions*. Elderly GBM patients often have a dismal prognosis. However, it is possible to identify a subgroup with favourable clinical and molecular features, who benefit from GTR and radio-chemotherapy with temozolomide. A comprehensive prognostic score is needed to guide treatment modality and predict the outcome.

## 1. Introduction

Glioblastoma (GBM) is the most frequent and aggressive primary brain tumour in adults, accounting for 14.5% of all brain tumours and 57.7% of gliomas [1]. The incidence of GBM increases with age. In fact, according to the Central Brain Tumour Registry of an American (CBTRUS) statistical report from the 2013–2017 period, the incidence of GBM is 3.23 per 100,000 people per year, reaching its peak in people 75–84-year-old (15.30 per 100,000 people per year) [1]. Moreover, as life expectancy is progressively increasing, the incidence of GBM in the elderly population will further rise in the future.

In this paper, we provide a case series of elderly patients with GBM from our Institution, in whom we assessed the clinical characteristics, feasibility and effectiveness of surgical resection, response to adjuvant treatments, and outcome, along with the impact of comorbidities and clinical status on survival. The aims of our study were: to investigate the impact of surgery and adjuvant treatment on survival; to assess whether comorbidities significantly impacted the choice of treatments and outcome; to assess how clinical characteristics and outcome were distributed across different classes of age; and to identify a subgroup of elderly patients with more favourable prognoses.

## 2. Patients and Methods

### 2.1. Retrospective Collection of Patients’ Data

From 2015 to 2020, we collected in a specific database all elderly (≥65 years) GBM patients who have been diagnosed and treated at the Division of Neuro-Oncology, University and City of Health and Science Hospital of Turin. 

The database collected all the information about clinical, pathological, molecular, and therapeutic characteristics of patients. From a dataset of 160 patients, we have selected only those patients who underwent a surgical procedure, with a histologically confirmed diagnosis of GBM *IDH*-wildtype. One hundred and thirty-five patients were available for the analysis.

### 2.2. Evaluation of Clinical Performance Status, Comorbidities, Molecular Characteristics, and Radiological Assessment

Clinical status was evaluated according to the Karnofsky Performance Status (KPS), both before surgery (at admission to the Neurosurgery Division), and post-operatively (by a dedicated Brain Tumour Board, which took place 30 days after surgery to decide the most appropriate adjuvant treatment modality). Comorbidity scores were assessed according to the Charlson Comorbidity Index (CCI) as follows: age (65–69 years: 2; 70–79 years: 3; ≥80 years: 4); myocardial infarction (no: 0; yes: 1); congestive heart failure (no: 0; yes: 1); peripheral vascular disease (no: 0; yes: 1); cerebrovascular disease (no: 0; yes: 1); dementia (no: 0; yes: 1); chronic pulmonary disease (no: 0; yes: 1); connective tissue disease (no: 0; yes: 1); peptidic ulcer disease (no: 0; yes: 1); liver disease (no: 0; mild: 1; moderate: 2); diabetes mellitus (none or diet-controlled: 0; uncomplicated: 1; end-organ damage: 2); hemiplegia (no: 0; yes: 2); chronic kidney failure (no: 0; yes: 2); solid tumour (no: 0; localised: 2; metastatic: 6); leukaemia (no: 0; yes: 2); lymphoma (no: 0; yes: 2); AIDS (no: 0; yes: 6). 

The presence of *IDH* mutation was ruled out in all cases by Sanger gene sequencing. The *MGMT* promoter methylation status was assessed by polymerase chain reaction (PCR).

Extent of resection (EOR) was defined as gross-total or subtotal based on the presence of a contrast-enhancing residual tumour on MRI after 24–72 h from surgery. Response Assessment in Neuro-Oncology (RANO) criteria were used to evaluate MRI response to treatments [2]. Treatment-related adverse events were defined according to the Common Terminology Criteria for Adverse Events (CTCAE) version 5.0.

### 2.3. Statistical Analysis

Baseline characteristics of patients included in the analysis are summarised using median and interquartile range (IQR) and percentages and frequencies (*n*, %). We adopted age at surgery as a surrogate of age at diagnosis. The observation period for progression-free survival (PFS) and overall survival (OS) started on the date of surgery until the date of recurrence or death, respectively, or until the last follow-up visit (censoring).

The distribution of characteristics between molecular subgroups were evaluated by the Mann–Whitney U test for continuous variables and the Chi-square test or Fisher’s exact test for categorical variables. Kaplan–Meier curves were drawn for PFS and OS and a Cox proportional hazard model was employed to estimate the crude and the multivariable-adjusted hazard ratios (HRs) with 95% confidence intervals (CIs) and to evaluate possible predictors of recurrence or survival. The proportional hazard assumption was also verified by graphical checks and formal tests based on Schoenfeld residuals. 

The analysis was performed by IBM SPSS Statistics v.28 software.

### 2.4. Ethical Approval

The project was approved by the Review Board of our Institution. Informed consent to collect and analyse clinical and pathological/molecular data was obtained from all subjects who were alive at the time of start of the study or from relatives in case of death of the subjects, according to ethic regulations for retrospective studies of the local Ethics Committee.

## 3. Results

### 3.1. Characteristics of Patient Population and Treatment Modalities

One hundred and thirty-five patients met the inclusion criteria. Median age was 71.0 years, with 32 patients (23.7%) being older than 75 years and 9 (6.7%) older than 80 years. The most common symptoms at onset were motor deficits (32, 23.7%), cognitive and behaviour disorders (30, 22.2%), speech disorders (23, 17.0%), and seizures (23, 17.0%). Overall, 47 patients (34.8%) had a history of seizures throughout the disease course. Most patients (127, 95.0%) had a baseline KPS ≥ 70. Sixty-eight (50.4%) patients suffered from at least one comorbidity. Median CCI was 3, and 61 (45.2%) patients had CCI > 3. No history of allergies was reported. Subtotal and gross-total resections accounted for 102 (75.6%) and 10 (7.4%) cases, respectively; biopsies were 23 (17.0%). After surgery, KPS was ≥70 in 105 (77.8%) patients. After surgery, 37 patients (27.4%) underwent hypofractioned (42 Gy in 3 weeks) concurrent radio-chemotherapy with temozolomide (TMZ), followed by adjuvant TMZ; 33 (24.4%) underwent long-course (60 Gy in 6 weeks) concurrent radio-chemotherapy with TMZ followed by adjuvant TMZ; radiotherapy (RT) alone (42 Gy), upfront chemotherapy with TMZ, and RT (42 Gy), followed by TMZ accounted for 14 (10.4%), 14 (10.4%), and 13 (9.6%) patients, respectively; 24 patients (17.8%) did not undergo any adjuvant treatments, in favour of best supportive care. All information of patient characteristics and treatment modalities are reported in the Table 1. Clinical characteristics did not differ according to the sex of patients (Appendix A).

### 3.2. Complications after Surgery and Correlation with the Extent of Resection (EOR)

Fifty-one (37.7%) patients presented clinical complications and/or neuroradiological impairment within one month of surgery before starting adjuvant treatments. In 28 cases, (20.7%) clinical complications occurred in the first week after surgery (see Table 2 for detailed information).

The occurrence of clinical complications after surgery was not associated with CCI or KPS at baseline. Moreover, there was no association between the entity of resection and occurrence of clinical complications, which was accounted for in 8/23 (34.8%) patients after biopsy, for 27/102 (26.5%) after subtotal resection, and for 1/10 (10.0%) patients after gross-total resection (*p* = 0.333).

### 3.3. Clinical Characteristics across Different Classes of Age

We evaluated the distribution of different clinical factors across different classes of age of our cohort (65–69; 70–74; 75–79; ≥80 years). First, older patients had a higher CCI (CCI score > 3). As age influences the final CCI score, we subtracted the value of age from the total CCI score to consider the weight of comorbidities only; even in this case, people older than 80 had a significantly higher index, being age-adjusted CCI > 3 in 3/52 (5.8%), 1/51 (2.0%), 0/23 and 2/9 (22.2%) among patients 65–69, 70–74, 75–79, and ≥80-year-old, respectively (*p* = 0.033). Higher age was also associated with higher incidence of clinical complications after surgery. Conversely, other clinical factors (such as incidence of seizures and multifocal presentation) as well as prevalence of *MGMT*p methylation did not differ among age subgroups. Regarding adjuvant treatments, 6-week concomitant RT/TMZ followed by adjuvant TMZ prevailed among patients younger than 75 years (31/103, 30.1% vs. 2/32, 6.2%), whereas those older than 75 years underwent more frequent 3-week RT followed by adjuvant TMZ (5/32, 15.6% vs. 8/103, 7.8%) (see Table 3). 

### 3.4. Clinical Characteristics of Patients Undergoing Different Adjuvant Treatments

Patients undergoing active treatments frequently displayed KPS ≥ 70 at baseline (107/111, 96.4% vs. 20/24, 83.3% of not-treated patients, *p* = 0.014) and after surgery (94/111, 84.7% vs. 11/24, 45.8% of not-treated patients, *p* < 0.001). Patients undergoing standard 6-week radio-chemotherapy with TMZ had a significantly younger median age than the other ones (68 vs. 71 years old within the general cohort, *p* = 0.005). Also, CCI > 3 prevailed among patients not undergoing concomitant radio-chemotherapy as compared to other strategies (31/65, 47.7% vs. 25/70, 35.0%, *p* < 0.001). *MGMT*p methylation was found in 19/37 (51.4%) patients undergoing hypofractioned 3-week RT/TMZ + TMZ, 19/33 (57.6%) and among those undergoing standard 6-week RT/TMZ + TMZ, and 7/14 (50.0%), 11/14 (78.6%) and 4/13 (30.8%) and those treated with RT alone, upfront TMZ, and RT followed by TMZ. However, this and all the remaining findings were not significant. The Table 4 shows the distribution of clinical characteristics of patients undergoing different adjuvant treatments and values of median progression-free survival and overall survival for each group.

### 3.5. Treatment-Related Adverse Events

Thirty-one out of 111 patients undergoing RT and/or TMZ reported adverse events (27.9%). Combined short or standard-course radio-chemotherapy had a higher rate of adverse events (17/37, 45.9%; 8/33, 24.2%) as compared to TMZ upfront (3/14, 21.4%), RT alone (2/14, 14.3%), and RT followed by TMZ (1/13, 7.7%) (*p* = 0.034). Temozolomide-related haematologic toxicity and nausea were the most common adverse events. All adverse events were grade 1–2 according to the CTCAE version 5. A list of main adverse events is contained in Table 5.

### 3.6. Radiological Response to 1st Line Treatments and Treatment Modalities at Progression

Best RANO response to first line treatment were complete response (CR) in 3/111 patients (2.7%), partial response (PR) in 8/111 (7.2%), stable disease (SD) in 23/111 (20.7%), and progressive disease (PD) in 77/111 (69.4%) (Table 1). Forty-one patients out of 135 (30.3%) underwent a second line treatment, which consisted of: lomustine/procarbazine (10/41, 24.4%), regorafenib (10/41, 24.4%), fotemustine (9/41, 22.0%), TMZ rechallenge (6/41, 14.6%), bevacizumab (4/41, 9.8%), salvage RT (2/41, 4.9%) (Table 1). 

### 3.7. Analysis of Survival and Prognostic Factors

#### 3.7.1. Progression-Free Survival and Overall Survival of the Whole Cohort

Median progression-free survival (mPFS) and overall survival (mOS) for the whole cohort were 8.0 months (5.6–10.4, 95% CI) and 10.5 months (9.0–11.9, 95% CI). Proportion of patients surviving at 6, 12, 18, and 24 months was 73.3%, 42.2%, 31.1%, and 22.2%, respectively (Table 1). 

#### 3.7.2. Outcome across Different Classes of Age and According to Clinical Performance Status and Comorbidities

Patients 65–69 years old had a mPFS of 7.7 months and a mOS of 11.0 months, whereas those 70–74 years old had a mPFS of 9.3 months and mOS of 10.1 months, and those 75–79 years old had a mPFS of 4.2 months and mOS of 8.8 months. However, differences in mPFS and mOS across different age classes were not significant (Table 3). Remarkably, the nine patients older than 80 years had quite a long mPFS (19.8 months) and mOS (28.7 months); this could be explained by the fact that all those nine patients received an active treatment after surgery, mostly RT/TMZ followed by adjuvant TMZ (3-week and 6-week schedule in 6/9 and 1/9, respectively), and upfront TMZ in two cases, which could represent a selection bias for a better outcome despite the advanced age. 

KPS ≥ 70 at baseline correlated with longer mPFS (8.4 vs. 1.0 months, *p* < 0.001) and mOS (10.7 vs. 4.0 months, *p* < 0.001). In the whole group, mPFS and mOS of patients with CCI ≤ or >3 did not differ significantly (Table 6). However, only among patients 65–69 years old, CCI > 3 correlated with worse mOS (4.4 vs. 12.3 months, *p* = 0.006) (Appendix A).

#### 3.7.3. Outcome According to Extent of Resection and Adjuvant Treatments

EOR displayed a significant impact on outcome, with mPFS and mOS being longer for patients undergoing gross-total resection (18.8 and 32.8 months, respectively) than subtotal resection (6.5 and 9.9 months) or biopsy (1.0 and 6.0 months) (*p* = 0.002 for mPFS and *p* < 0.001 for mOS models, respectively). Clinical complications after surgery did not significantly affect mPFS (8.2 vs. 7.9 months, *p* = 0.406) and mOS (10.7 vs. 10.5 months, *p* = 0.675). 

Among patients undergoing adjuvant treatments (111, 82.2%), those who underwent 6-week RT/TMZ followed by TMZ had the longest mPFS (18.8 months) and mOS (28.4 months), followed by those who underwent RT followed + TMZ (mPFS 12.7 months; mOS 23.3 months) and 3-week combined RT/TMZ + TMZ (mPFS 10.4 months; mOS 16.0 months). Ultimately, mPFS and mOS of patients who underwent RT alone after surgery were 3.9 and 6.0 months, respectively, whereas in patients who were treated with TMZ alone they were 3.3 and 8.3 months, respectively. 

Among patients who underwent concomitant RT/TMZ + TMZ who displayed more frequent treatment-related adverse events (Table 5), and those who developed toxicity from treatments did not show worse mPFS (10.3 vs. 13.0, *p* = 0.385) or mOS (19.1 vs. 19.2 months, *p* = 0.769).

Median progression-free and overall survival according to EOR and adjuvant treatment are reported in Table 7.

#### 3.7.4. Outcome According to *MGMT*p Status within Groups of Different Treatment Modalities

Within the whole cohort, *MGMT*p methylation was associated with significantly better mPFS (11.7 vs. 6.0 months, *p* < 0.001) and mOS (19.0 vs. 9.8 months, *p* < 0.001).

We also evaluated mPFS and mOS of patients undergoing different adjuvant treatment modalities after surgery according to *MGMT*p status. The presence of *MGMT*p methylation correlated with a significantly longer mPFS and mOS among patients undergoing 6-week RT/TMZ + TMZ (mPFS: 20.8 vs. 9.6 months, *p* < 0.001; mOS: 44.8 vs. 10.7 months, *p* < 0.001) and 3-week RT/TMZ (mPFS: 19.5 vs. 7.6 months, *p* < 0.001; mOS: 27.1 vs. 12.9 months, *p* = 0.002). A non-significant trend for better mOS was also seen among patients with *MGMT*p methylation treated with upfront TMZ (8.3 vs. 4.6 months, *p* = 0.090); conversely, among patients undergoing RT alone or RT followed by TMZ, the presence of *MGMT*p methylation did not significantly affect the outcome (Table 8).

#### 3.7.5. Multivariable Analysis of Prognostic Factors on Progression-Free and Overall Survival

In a multivariable analysis of the whole cohort, factors that significantly reduced the risk of progression in a multivariable analysis were: *MGMT*p methylation, large surgical resection, and adjuvant radio-chemotherapy; factors that significantly reduced the risk of death were: KPS at baseline ≥ 70, *MGMT*p methylation, large surgical resection, adjuvant radio-chemotherapy, and second line treatments (Table 9).

To conclude, the subgroup of patients who displayed the best outcome were those with KPS at baseline ≥ 70, with *MGMT*p methylation, who underwent gross-total resection and adjuvant radio-chemotherapy with TMZ: in this subgroup, mPFS was 35.5 months and mOS was not reached. 

## 4. Discussion

Our single, large institutional study on elderly GBM confirms that glioblastoma in the elderly population (≥65 years) is a highly aggressive tumour, with mPFS and mOS being 8.0 months and 10.5 months, respectively. However, our data show that there is still some heterogeneity within this cohort of patients, giving rise to still unanswered issues.

First, which is the most appropriate threshold to define elderly age is not clear, as this population may be further stratified by different classes of age based on clinical characteristics and outcome. In fact, a non-significant (but still noteworthy) trend for worse survival in older patients was seen in our cohort, when comparing 65–69 year-old patients (mPFS 7.7 months; mOS 11.0 months) to 70–74 year-olds (mPFS 9.3 months; mOS 10.7 months), and to 75–79 year-olds (mPFS 4.2 months; mOS 8.8 months) (Table 3). In our series, a higher prevalence of comorbidities and an increased risk of clinical complications after surgery were significantly associated with higher age, which explains the poorer outcome of the oldest patients within the series (Table 3). 

Second, which is the most appropriate comorbidity score in elderly patients with GBM has not been established thus far. The Charlson Comorbidity Index was developed to predict the 10-year mortality of patients, scoring from 1 to 6 depending on the risk of death from comorbidities, and it is uncertain whether such a score may be adequate in glioblastoma patients with dramatically short survival [3]. In our series, the CCI was not a reliable predictor of outcome, except for the youngest patients of the cohort (65–69 years old), who displayed a poorer survival with CCI > 3 (Appendix A). This was probably due to the prominent weight of age rather than comorbidities in determining the CCI score of our patients. In fact, the number of patients without comorbidities was considerable (67/135, 49.6%); in these cases, age was the only factor determining the CCI score (Table 1).

A strong correlation between CCI and outcome was not found in previous studies involving elderly GBM patients. In a study on 146 patients with glioblastoma (where 56 patients older than 65 years were included), CCI < 2 did not correlate with outcome. However, only 8/56 elderly patients had a CCI > 2, which made this association uncertain [4]. Conversely, in another small study on 35 patients older than 65 years treated with radio-chemotherapy, age-adjusted CCI correlated with prognosis: patients with CCI < 3 (22/35, 62.8%) had a mOS of 22 months, whereas those with CCI ≥ 3 (13/35, 37.2%) had a mOS of 10 months [5]. However, overall survival of elderly patients with CCI < 3 was longer than what has been reported in previous series; second, the number of patients with CCI ≥ 3 was smaller than usually seen in similar series; third, other clinical and molecular factors (i.e., *MGMT*p methylation), which might have clarified these issues, were not investigated [5]. In another study on 233 elderly patients, CCI > 3 correlated with worse mPFS and mOS [6]. However, patients had a lower median age than those of our series (62 years vs. 71 years), with similar prevalence of patients with CCI > 3 (about 46% vs. 45.6%) and same median CCI score (3, as in our case). This resembles what we observed in our cohort, where CCI > 3 retained a prognostic importance only in the 65–69-year-old class of age; this was probably due to the impact of comorbidities on outcome in younger rather than older patients who have an independent risk for dismal prognosis and very short survival (Table 6; Appendix A).

In our study, KPS influenced the choice of treatments (Table 4) and correlated with outcome. Overall, most patients had KPS ≥ 70 at diagnosis. However, patients not undergoing surgery or adjuvant treatments had worse KPS (Table 4). In the multivariable analysis (Table 9), KPS ≥ 70 at baseline favourably impacted the overall survival of the whole cohort (OR 0.487, 0.214–0.908 95% CI). Therefore, similar to other series [7,8,9], KPS was effective in identifying patients with poorer outcomes. 

However, other factors had a heavy impact on prognosis. In our series we suggested that surgical resection, as large as possible, represents one of the strongest prognostic factors (Table 7 and Table 9). The prognostic role of the extent of surgery among elderly patients with GBM has long been debated, due to concern for a higher risk of surgical-related complications in such a frail population. Evidence of feasibility and safety of extended resections in elderly GBM patients mainly derives from small case series, whereas data from phase III trials are lacking. Preliminary data about the superiority of open craniotomy over stereotactic biopsy in patients aged more than 65 initially came from small retrospective or prospective studies on 30–40 patients [9,10]. Then, a larger retrospective trial on 142 elderly patients with newly diagnosed GBM established the superiority of extended resection over biopsy in terms of overall survival (13.0 months vs. 4.0 months, *p* < 0.001) [11]. Finally, data from randomised phase III trials NOA-08 and Nordic confirmed that surgical resection is superior to biopsy alone [12,13]. In our study, patients undergoing extended surgery had longer mPFS and mOS (Table 7). Moreover, EOR retained a significant impact on prognosis, being mOS after gross-total resection (GTR) of 32.8 months vs. 9.9 after subtotal resection and 6.0 after biopsy (Table 7). Survival after GTR was remarkably long, which was also explained by the fact that patients undergoing GTR were more frequently treated with combined radio-chemotherapy regimens (Table 4). Additionally, the incidence of clinical complications was not higher in case of GTR, thus proving that GTR was a safe option for elderly patients. As in the multivariable analysis, the EOR retains an independently favourable prognostic role (Table 9), and our study confirms that gross-total resection (GTR), if feasible, is the best option for GBM patients [14,15,16], regardless of molecular status [17]. 

Adjuvant treatments after surgical resection dramatically impacted the outcome. In our study, we confirmed that combined RT/TMZ followed by TMZ provides the longest survivals in the whole cohort, regardless of *MGMT*p methylation status (Table 7). However, among patients with *MGMT*p methylation, survival was longer with RT/TMZ followed by adjuvant TMZ, whereas among those without *MGMT*p methylation combined radio-chemotherapy did not provide a significant benefit (Table 7). These data are consistent with the findings of Perry et al. [18], even if, in our case, the 3-week RT/TMZ regimen was superior to the standard 6-week RT/TMZ only among *MGMT*p-unmethylated patients, whereas among *MGMT*p-methylated patients 6-week combined RT/TMZ was still associated with longer overall survival (Table 8).

Lastly, similar to the findings of the Nordic trial [13], in our study, *MGMT*p-unmethylated patients had a better survival when treated with hypofractioned RT, as compared to upfront CT with TMZ (16.1, 12.9, 6.0, and 4.6 months in the 3-week RT + TMZ, 3-week RT/TMZ + TMZ, hypofractioned RT alone, and upfront TMZ groups, respectively–Table 8), which suggests that the use of combined radio-chemotherapy schedule in this molecular subgroup might be avoided to improve tolerability.

In Table 10, a list of the main studies on elderly GBM patients that have been published from the introduction of Stupp regimen (2005) is reported, with a particular focus on patient and tumour characteristics, post-surgical complications, treatment modalities, and outcome [19,20,21,22,23,24,25,26,27,28,29,30,31,32,33]. As compared to other studies, we could identify the presence of different subgroups of patients with different outcomes according to clinical and molecular characteristics (i.e., *MGMT*p status), extent of resection, and adjuvant radio-chemotherapy. Even if the general prognosis of the whole cohort remains poor (similar to other series), we recognised a subgroup with better outcomes, with remarkably longer mPFS and mOS, characterised by the presence of *MGMT*p methylation, and after gross-total resection and adjuvant radio-chemotherapy. The implementation of a comprehensive geriatric score would be of primary importance to suggest the best treatment modality to improve survival and avoid treatment-related adverse events among elderly people with frail conditions who may not tolerate aggressive treatments [34].

## 5. Conclusions

Elderly GBM patients are generally characterised by limited response to treatment and poor outcomes. However, in our study we identify a subgroup of patients with favourable clinical and molecular features who significantly benefit from large surgical resection and radio-chemotherapy. A comprehensive evaluation of clinical and molecular characteristics of elderly GBM patients is essential to choose the best treatment modality and predict the outcome, in order to increase survival, reduce toxicity, and improve quality of life.

## Figures and Tables

**Table 1 brainsci-12-00632-t001:** Patient Characteristics (*n* = 135).

Median Age, Years (*n*, %)	71.0 (68.0–74.0 IQR)	
65–69 years	52	38.5%
70–74 years	51	37.8%
75–79 years	23	17.0%
≥80 years	9	6.7%
Sex (*n*, %)		
Male	87	64.4%
Female	48	35.6%
Symptoms at Onset (*n*, %)		
Motor Deficit	32	23.7%
Cognitive/Behaviour Disorder	30	22.2%
Speech Disorder	23	17.0%
Seizures	23	17.0%
Headache	11	8.1%
Visual Disorder	7	5.2%
Somato-sensorial Deficit	4	3.0%
Incidental Finding	5	3.7%
History of Seizures (at any time)	47	34.8%
Karnofsky Performance Status (KPS) at Baseline (*n*, %)		
≥90	22	16.3%
80	54	40.0%
70	51	37.8%
60	7	5.2%
50	1	0.7%
Comorbidities (*n*, %)		
Any Comorbidity	68	50.4%
Diabetes (*uncomplicated*)	29	21.5%
Systemic Tumour (*localised*)	21	15.6%
Chronic Pulmonary Disease	11	8.1%
Myocardial Infarction	11	8.1%
Peripheral Vascular Disease	10	7.4%
Congestive Heart Failure	4	3.0%
Cerebrovascular Disease	4	3.0%
Dementia	1	0.7%
Connective Tissue Disease	1	0.7%
Peptidic Ulcer Disease	3	2.2%
Liver Disease (*mild*)	3	2.2%
Hemiplegia	1	0.7%
Chronic Kidney Disease	none	
Diabetes (*end-organ damage*)	none	
Leukaemia	none	
Lymphoma	none	
Acquired immunodeficiency syndrome (AIDS)	none	
Charlson Comorbidity Index (CCI)		
Median CCI	3 (3–4 IQR)	
≤3 (*n*, %)	74	54.8%
>3 (*n*, %)	61	45.2%
Multifocal Tumour	16	11.8%
*MGMT*p Methylation Status (*n*, %)		
Methylated	70	51.9%
Unmethylated	58	43.0%
Unknown	7	5.2%
Extent of Resection (EOR) (*n*, %)		
Subtotal	102	75.6%
Gross-total	10	7.4%
Biopsy	23	17.0%
Karnofsky Performance Status (KPS) after Surgery (*n*, %)		
≥90	26	19.3%
80	40	29.6%
70	39	28.9%
60	20	14.8%
50	8	5.9%
40	2	1.5%
Management after Surgery (*n*, %)		
3-weeks RT/TMZ + TMZ	37	27.4%
6-weeks RT/TMZ + TMZ	33	24.4%
Best supportive care	24	17.8%
RT alone	14	10.4%
TMZ upfront	14	10.4%
RT + TMZ	13	9.6%
RANO Response to First-Line Treatment (*n* = 111) (*n*, %)		
Complete Response	3	2.7%
Partial Response	8	7.2%
Stable Disease	23	20.7%
Progressive Disease	77	69.4%
Management at Progression (*n* = 41) (*n*, %)		
Lomustine/procarbazine	10	24.4%
Regorafenib	10	24.4%
Fotemustine	9	22.0%
TMZ rechallenge	6	14.6%
Bevacizumab	4	9.8%
Salvage RT	2	4.9%
Median Progression-Free Survival (months, 95% CI)	8.0 months (5.6–10.4)	
Progression-free patients (*n*, %)		
At 6 months (*n*, %)	69	51.1%
At 12 months (*n*, %)	40	29.6%
At 18 months (*n*, %)	28	20.7%
At 24 months (*n*, %)	18	13.3%
Median Overall Survival (months, 95% CI)	10.5 (9.0–11.9)	
Surviving patients (*n*, %)		
At 6 months (*n*, %)	99	73.3%
At 12 months (*n*, %)	57	42.2%
At 18 months (*n*, %)	42	31.1%
At 24 months (*n*, %)	30	22.2%

Abbreviations: IQR, interquartile range; *MGMT*, O(6)-methylguanyl DNA methyltransferase promoter; RANO, Radiological Assessment in Neuro-Oncology; RT, radiotherapy; TMZ, temozolomide.

**Table 2 brainsci-12-00632-t002:** Clinical Complications within One Month from Surgery.

Complications/Clinical Impairment after Surgery (*n*, %)		
Early (within one week)	28	20.7%
Delayed (>one week, within one month)	23	17.0%
Within the 1st Week from Surgery (*n*, %)		
Status Epilepticus	4	14.3%
Stroke	4	14.3%
Admission to Intensive Care Unit (ICU)	3	10.7%
Systemic Infection	3	10.7%
Neurological Impairment	2	7.1%
Delirium	2	7.1%
Deep Venous Thrombosys (DVT)	2	7.1%
Anaemia	2	7.1%
Pulmonary Embolism (PE)	2	7.1%
Bowel Perforation	1	3.6%
Acute Heart Failure	1	3.6%
Severe Hyperglycaemia	1	3.6%
Iatrogenous Meningitis	1	3.6%
After One Week from Surgery, within One Month (*n*, %)		
Neurological Impairment	6	26.1%
Pulmonary Embolism (PE)	4	17.4%
Diabetes	3	13.0%
Systemic Infection	2	8.7%
Trauma	2	8.7%
Meningitis	2	8.7%
Seizures	1	4.3%
Bowel Obstruction	1	4.3%
Subdural Haematoma	1	4.3%
Anaemia	1	4.3%

**Table 3 brainsci-12-00632-t003:** Distribution of clinical characteristics within different classes of age.

	65–69	70–74	75–79	≥80	*p* Value
Seizures	18/52 (34.6%)	17/51 (33.3%)	10/23 (43.5%)	2/9 (22.2%)	0.696
Multifocal GBM	6/52 (11.5%)	5/51 (9.8%)	2/23 (8.7%)	3/9 (33.3%)	0.221
*MGMT*p methylation	27/51 (52.9%)	26/46 (56.5%)	11/22 (50.0%)	6/9 (66.7%)	0.840
CCI > 3	11/52 (21.2%)	25/51 (49.0%)	11/23 (47.8%)	9/9 (100%)	<0.001
Age-adjusted CCI > 3 *	3/52 (5.8%)	1/55 (2.0%)	0/23 (0.0%)	2/9 (22.2%)	0.033
KPS before surgery ≥ 70	50/52 (96.2%)	47/51 (92.2%)	22/23 (95.7%)	8/9 (88.9%)	0.735
KPS after surgery ≥ 70	41/52 (78.8%)	37/51 (72.5%)	19/23 (82.6%)	8/9 (88.9%)	0.616
Complications after surgery	16/52 (30.8%)	16/51 (31.4%)	0/23 (0.0%)	4/9 (44.4%)	0.013
Adjuvant Treatments					
6-week RT/TMZ + TMZ	19/52 (36.5%)	12/51 (23.5%)	1/23 (4.3%)	1/9 (11.1%)	0.007
3-week RT/TMZ + TMZ	15/52 (28.8%)	11/51 (21.6%)	5/23 (21.7%)	6/9 (66.7%)	
RT + TMZ	1/52 (1.9%)	7/51 (13.7%)	5/23 (21.7%)	0/9 (0.0%)	
RT alone	2/52 (3.8%)	8/51 (15.7%)	4/23 (17.4%)	0/9 (0.0%)	
TMZ upfront	6/52 (11.5%)	4/51 (7.8%)	2/23 (8.7%)	2/9 (22.2%)	
Palliative care	9/52 (17.3%)	9/51 (17.6%)	6/23 (26.1%)	0/9 (0.0%)	
mPFS (months, 95% CI)	7.7 (5.1–10.4)	9.3 (4.2–14.5)	4.2 (0.1–8.9)	19.8 (10.1–29.5)	0.060
mOS (months, 95% CI)	11.0 (7.2–14.9)	10.1 (7.9–12.1)	8.8 (7.1–10.5)	28.7 (17.9–39.2)	0.128

Abbreviations: CCI, Charlson Comorbidity Index; GBM, glioblastoma; KPS, Karnofsky Performance Status; *MGMT*, O(6)-methylguanyl DNA methyltransferase promoter; mOS, median overall survival; mPFS, median progression-free survival; RT, radiotherapy; TMZ, temozolomide. * CCI was modified by subtracting the value attributed to age from the total score, to make comorbidities the only factors included in the index.

**Table 4 brainsci-12-00632-t004:** Characteristics of patients within different groups of treatments.

	3-Week RT/TMZ + TMZ	6-Week RT/TMZ + TMZ	RT Alone	TMZ Upfront	RT + TMZ	Palliation	
Total	37	33	14	14	13	24	
	27.4%	24.4%	10.4%	10.4%	9.6%	17.8%	
Median age (years)	71.0	68.0	71.0	72.0	73.0	72.0	*p* = 0.005
CCI > 3	19	6	11	5	8	7	*p* < 0.001
	51.4%	18.2%	78.6%	35.7%	61.5%	29.2%	
Multifocal tumour	6	5	1	4	0	0	*p* = 0.074
	16.2%	15.2%	7.1%	28.6%	0.0%	0.0%	
*MGMT*p methylation	19	19	7	11	4	10	*p* = 0.178
	51.4%	57.6%	50.0%	78.6%	30.8%	41.7%	
Gross-total resection	6	9	1	0	4	3	*p* = 0.136
	16.2%	27.3%	7.1%	0.0%	30.8%	12.5%	
KPS ≥ 70 at baseline	36	33	13	12	13	20	*p* = 0.066
	97.3%	100.0%	92.9%	85.7%	100.0%	83.3%	
KPS ≥ 70 after surgery	34	33	10	5	12	11	*p* < 0.001
	91.9%	100.0%	71.4%	35.7%	92.3%	45.8%	
Clinical complication after surgery	13	9	4	3	2	5	*p* = 0.723
	35.1%	27.3%	28.6%	21.4%	15.4%	20.8%	
mPFS (months, 95% CI)	10.4 (8.6–12.0)	18.8 (9.6–28.0)	3.9 (2.7–5.0)	3.3 (2.4–4.1)	12.7 (9.4–15.9)	1.0 (0.9–1.1)	*p* < 0.001
mOS (months, 95% CI))	16.0 (8.4–23.7)	28.4 (15.9–40.8)	6.0 (4.8–7.3)	8.3 (6.6–10.0)	23.3 (8.6–37.9)	3.4 (3.0–3.7)	*p* < 0.001

Abbreviations: CI, coefficient interval; CCI, Charlson Comorbidity Index; KPS, Karnofsky Performance Status; *MGMT*, O(6)-methylguanyl DNA methyltransferase promoter; mPFS, median progression-free survival; mOS, median overall survival; RT, radiotherapy; TMZ, temozolomide. Please note: “palliation” refers to patients not undergoing adjuvant treatments after surgery.

**Table 5 brainsci-12-00632-t005:** Treatment-Related Adverse Events.

	3-Week RT/TMZ + TMZ	6-Week RT/TMZ + TMZ	TMZ Upfront	RT Alone	RT + TMZ	Total
Any toxicity	17/37 (45.9%)	8/33 (24.2%)	3/14 (21.4%)	2/14 (14.3%)	1/13 (7.7%)	31/111 (27.9%)
Haematologic	7/37 (18.9%)	2/33 (6.1%)	2/14 (14.3%)	2/14 (14.3%)	/	13/111 (11.7%)
Nausea	7/37 (18.9%)	3/33 (9.1%)	/	/	1/13 (7.7%)	11/111 (9.9%)
Fatigue	1/37 (2.7%)	3/33 (9.1%)	1/14 (7.1%)	/	/	5/111 (4.5%)
Secondary parkinsonism	2/37 (5.4%)	/	/	/	/	2/111 (1.8%)

Abbreviations: RT, radiotherapy; TMZ, temozolomide.

**Table 6 brainsci-12-00632-t006:** Progression-free Survival and Overall Survival according to the Karnofsky Performance Status and Charlson Comorbidity Index.

	Progression-Free Survival		Overall Survival	
	Months, 95% CI	*p* Value	Months, 95% CI	*p* Value
KPS				
KPS ≥ 70	8.4 (5.9–10.9)	0.002	10.7 (8.2–13.2)	<0.001
KPS < 70	1.0 (0.9–1.1)		4.0 (2.9–5.1)	
Age-adjusted CCI *				
CCI ≤ 3	8.1 (6.0–10.3)	0.310	10.7 (8.1–13.2)	0.386
CCI > 3	2.5 (0.1–5.6)		4.5 (0.1–10.2)	

Abbreviations: CI, coefficient interval; CCI, Charlson Comorbidity Index; KPS, Karnofsky Performance Status. * CCI was modified by subtracting the value attributed to age from the total score, to make comorbidities the only factors included in the index.

**Table 7 brainsci-12-00632-t007:** Progression-free Survival and Overall Survival according to the Extent of Resection and Adjuvant Treatment Modalities.

	Progression-Free Survival		Overall Survival	
	Months, 95% CI	*p* Value	Months, 95% CI	*p* Value
Extent of resection				
Gross-total resection	18.8 (8.3–29.4)	0.002	32.8 (12.2–53.4)	<0.001
Subtotal resection	6.5 (3.3–9.7)		9.9 (8.4–11.5)	
Biopsy	1.0 (0.1–5.0)		6.0 (1.8–10.2)	
Adjuvant Treatment				
6-week RT/TMZ + TMZ	18.8 (9.6–28.1)	<0.001	28.4 (15.9–40.8)	<0.001
3-week RT/TMZ + TMZ	10.4 (8.6–12.0)		16.0 (8.4–23.7)	
RT + TMZ	12.7 (9.4–15.9)		23.3 (8.6–37.9)	
RT alone	3.9 (2.8–5.0)		6.0 (4.8–7.3)	
Upfront TMZ	3.3 (2.5–4.1)		8.3 (6.6–10.0)	

Abbreviations: CI, coefficient interval; RT, radiotherapy; TMZ, temozolomide.

**Table 8 brainsci-12-00632-t008:** Progression-free Survival and Overall Survival across Different Groups of Adjuvant Treatment According to *MGMT*p methylation status.

Treatment	Progression-Free Survival (Months, 95% CI)			Overall Survival (Months, 95% CI)		
	*MGMT*p Methylated	*MGMT*p Non-Methylated	*p* Value	*MGMT*p Methylated	*MGMT*p Non-Methylated	*p* Value
All patients (regardless of treatment)	11.7 (5.7–17.7)	6.0 (5.0–7.0)	<0.001	19.0 (6.7–31.4)	9.8 (8.6–11.0)	<0.001
6-week RT/TMZ + TMZ	20.8 (6.7–34.9)	9.6 (5.8–13.4)	<0.001	44.8 (24.6–65.0)	10.7 (9.7–11.7)	<0.001
3-week RT/TMZ + TMZ	19.5 (9.7–29.4)	7.6 (4.8–10.5)	<0.001	27.1 (17.8–36.4)	12.9 (10.3–15.6)	0.002
RT + TMZ	14.3 (5.6–22.9)	12.3 (8.3–16.5)	0.819	23.3 (0.1–47.7)	16.1 (0.1–39.3)	0.874
RT alone	3.7 (2.3–5.0)	4.4 (2.6–6.1)	0.152	6.6 (4.8–8.4)	6.0 (3.7–8.4)	0.804
Upfront TMZ	3.8 (0.3–7.3)	3.2 (1.6–4.7)	0.169	8.3 (6.9–9.7)	4.7 (3.4–5.9)	0.090

Abbreviations: CI, coefficient interval; *MGMT*p, O(6)-methylguanyl DNA methyltransferase promoter; RT, radiotherapy; TMZ, temozolomide.

**Table 9 brainsci-12-00632-t009:** Multivariable Analysis on Progression-Free Survival and Overall Survival.

	Progression-Free Survival			
	HR	95.0% CI		*p* Value
		Lower	Upper	
Age	0.965	0.915	1.018	0.190
CCI > 3	1.092	0.683	1.747	0.712
KPS at baseline ≥ 70	0.519	0.226	1.193	0.123
Extent of resection				
STR vs. GTR	2.834	1.591	5.049	<0.001
Biopsy vs. GTR	4.466	1.854	10.763	0.001
*MGMT*p methylation	0.561	0.366	0.860	0.008
Combined RT/TMZ	0.302	0.192	0.472	<0.001
	**Overall Survival**			
	HR	95.0% CI		*p* value
		Lower	Upper	
Age	0.963	0.911	1.017	0.178
CCI > 3	0.947	0.584	1.536	0.825
KPS at baseline ≥ 70	0.487	0.214	0.908	0.048
Extent of resection				
STR vs. GTR	2.539	1.317	4.895	0.005
Biopsy vs. GTR	4.194	1.642	10.712	0.003
*MGMT*p methylation	0.569	0.370	0.877	0.011
Combined RT/TMZ	0.368	0.233	0.580	<0.001
Second line treatment	0.460	0.285	0.742	0.001

Abbreviations: CI, coefficient interval; CCI, Charlson Comorbidity Index; GTR, gross-total resection; HR, hazard ratio; KPS, Karnofsky Performance Status; *MGMT*, O(6)-methylguanyl DNA methyltransferase promoter; RT, radiotherapy; STR, subtotal resection; TMZ, temozolomide.

**Table 10 brainsci-12-00632-t010:** Main clinical studies on elderly GBM patients investigating the role of EOR and adjuvant treatment on outcome.

Study	Pts.	Age	KPS	Comorbidity	*MGMT*p Methylation	EOR	Postoperative Deficits / Complications	Adjuvant Treatment	PFS (Months)	OS (Months)
Kleinschmidt, 2005 [19]	20	≥75	n.a.	n.a.	n.a.	Resection (12) Biopsy (6)	n.a.	RT alone (6) RT + CT (2) CT alone (1) BSC (11)	n.a	All patients: 4.6
Combs, 2008 [20]	43	≥65	26/43 ≥ 70 (60.4%)	n.a.	n.a.	STR (17) Biopsy (14) GTR (12)	n.a.	RT/TMZ (43) + TMZ (5)	n.a.	All patients: 11.0 GTR: 18.0 STR: 16.0 Biopsy: 6.0
Sijben, 2008 [21]	39	≥65	All ≥ 60	n.a.	29/35 (82.8%)	Resection (26) Biopsy (13)	n.a	RT alone (20) RT/TMZ + adjuvant TMZ (19)	Concomitant RT/TMZ + TMZ: 6.0RT alone: 4.1 Resection: 5.2 Biopsy: 5.0 *MGMT*p-met: 4.5 *MGMT*p-unmet: 5.5	RT/TMZ + TMZ: 8.5 RT alone: 5.2 Resection: 8.5 Biopsy: 5.0 *MGMT*p-met: 7.4 *MGMT*p-unmet: 7.3
Gerstein, 2010 [22]	51	≥65	44/51 ≥ 70 (86.3%)	n.a.	n.a.	Biopsy (23) STR (15) GTR (13)	n.a.	RT/TMZ (51) + TMZ (10)	All patients: 5.5 GTR: 9.5 STR: 4.17 Biopsy: 4.73	All patients: 11.5 GTR: 27.4 STR: 15.5 Biopsy: 7.89
Kimple, 2010 [23]	30	≥70	All ≥ 60	n.a.	n.a.	Biopsy (14) GTR (9) STR (7)	n.a.	RT/TMZ (14) + TMZ (9) BSC (13) RT alone (4)	n.a.	All patients: 20.6 wks RT/TMZ + TMZ: 50.5 wks RT alone: 28.2 wks BSC: 8.4 wks GTR: 26 wks Biopsy: 20.6 STR: 13.2
Lai, 2010 [24]	1355	≥65	n.a.	n.a.	n.a.	GTR (574) STR (485) Biopsy (296)	n.a.	RT (all patients) + CT (370)	n.a.	GTR: 9.3 STR: 8 Biopsy: 5.6
Laigle-Donadey, 2010 [25]	39	≥70	All ≥ 70	n.a.	13/28 (46.4%)	Biopsy (21) STR (14) GTR (3)	n.a.	Up-front TMZ	All patients: 20 wks No relationship with KPS, *MGMT*p status, EOR.	All patients: 36 wks No relationship with KPS, *MGMT*p status, EOR.
Chaichana, 2011 [9]	80	≥65	All ≥80	n.a.	n.a.	Biopsy (40) STR (25) GTR (15)	After resection: motor deficit (5): language deficit (1); infection (1) After biopsy: motor deficit (2); language deficit (1); infection (1); death (1)	RT (64) TMZ (8)	n.a.	All patients: 4.9 Resection: 5.7 Biopsy: 4.0
Ewelt, 2011 [26]	103	≥65	66/103 ≥ 70 (64.0%)	n.a.	n.a.	Biopsy (43) STR (37) GTR (23)	n.a.	RT (37) RT + TMZ (35) BSC (31)	n.a.	Age <75 years: 5.8 Age ≥75 years: 2.5 KPS <70: 2.4 KPS ≥70: 6.5 Biopsy (2.2) STR (7.0) GTR (13.9) RT + TMZ: 15.0 RT: 4.5 No adjuvant treatment: 1.8 GTR + RT + TMZ: 18.6 STR + RT + TMZ: 13.6 Biopsy + RT + TMZ: 7.3
Kushnir, 2011 [27]	68	≥65	All ≥ 65	n.a.	n.a.	Resection (42) Biopsy (26)	n.a.	RT + CT (27) RT alone (8) BSC (5)	n.a.	Resection: 11.1 Biopsy: 4.93 No adjuvant treatment: 3.8 RT alone: 9.47 RT + CT: 12.1
Hashem, 2012 [28]	20	≥60	13/20 ≥ 70 (65.0%)	n.a.	n.a.	Biopsy (10) STR (8) GTR (2)	n.a.	RT/TMZ (16) RT alone (1)	n.a.	All patients: 12.1 Biopsy: 8.26 STR: 15.41 GTR: 21.25
Tanaka, 2013 [29]	105	≥65	All ≥ 70	Cancer (23) CAD (21) DM (13) Hypertension (53) Hyperlipemia (47)	n.a.	Biopsy (52) Resection (53)	After biopsy: 16 AfterrResection: 10	RT/CT (41) RT alone (23) TMZ alone (1) BSC (19)	All patients: 3.5 Maximal safe resection + RT + CT: 8 Factors associated with shorter PFS: low KPS score, deep lesions, multifocal lesions, biopsy only, new persistent postoperative focal deficit, lack of adjuvant treatment No impact of comorbidities on PFS.	All patients: 5.5 Maximal safe resection + RT + CT: 12.5 Factors associated with shorter OS: same as PFS. No impact of comorbidities on OS.
Hoffermann, 2014 [30]	124	≥65	Mean: 70	n.a.	n.a.	Biopsy (17) STR (62) GTR (35)	After STR: 42.9% After GTR: 28.6% After biopsy: 7.4%	RT/TMZ + TMZ (39) RT alone (7) CT alone (6) RT + TMZ (6) BSC (45)	n.a.	All patients: 6.0 Biopsy: 4 STR: 9 GTR: 15 RT/TMZ + TMZ: 18.0 RT alone: 4.0 CT alone: 8.0 RT + TMZ: 15.0 BSC: 2.0 No impact of post-surgical complications on OS
Lombardi, 2015 [31]	237	≥65	≥60	n.a.	83/151 (54.9%)	STR (63) GTR (174)	n.a.	40 Gy RT/TMZ + TMZ (71) 60 Gy RT/TMZ + TMZ (166)	All patients: 11.3	All patients: 17.3 STR: 16.1 GTR: 17.7 60 Gy RT: 19.4 40 Gy RT: 13.8 *MGMT*p-met: 21.2 *MGMT*p-unmet: 13.6 No impact of ECOG on OS
Karsy, 2018 [32]	82	≥75	Median = 80	Hypertesion (36) CAD (21) Cancer (11) DM (11) DVT (7)	Found in 12 (14.6%; unknown in 61, 74.3%)	Biopsy (18) STR (33) GTR (19)	Biopsy (2) STR (5) GTR (2)	RT (32) TMZ (22) Bevacizumab (7) Other (4) BSC (17)	n.a.	Biopsy: 3.7 STR: 5 GTR: 12.1 No benefit from EOR in patients with surgical complications
Pessina, 2018 [33]	178	≥65	142/178 ≥ 70 (79.9%)	n.a.	103/178 (57.9%)	Biopsy (45) STR (62) GTR (63) CR (8)	Biopsy (4) STR (4) GTR (3) CR (0)	RT/TMZ (149) RT alone (29) Adjuvant TMZ (132)	All patients: 8.9	All patients: 12.2 Biopsy: 8.1 STR: 11.9 GTR: 15.1 CR: 24.5
Bruno et al., present study	135	≥65	127/135≥ 70 (94.0%)	CCI > 3 61/135 (45.2%)	70/135 (51.9%)	STR (102) GTR (10) Biopsy (23)	Biopsy (8) STR (27) GTR (1)	3-week RT/TMZ + TMZ (37) 6-week RT/TMZ + TMZ (33) RT alone (14) TMZ upfront (14) RT + TMZ (13) BSC (24)	All patients: 8.0 GTR: 18.8 STR: 6.5 Biopsy: 1.0 6-week RT/TMZ + TMZ: 18.8 RT + TMZ: 12.7 3-week RT/TMZ + TMZ: 10.4 RT alone: 3.9 TMZ upfront: 3.3 In *MGMT*p-met vs. unmet pts: 6-week RT/TMZ + TMZ: 20.8 vs. 9.6 3-week RT/TMZ + TMZ: 19.5 vs. 7.6	All patients: 10.5 GTR: 32.8 STR: 9.9 Biopsy: 6.0 6-week RT/TMZ + TMZ: 28.4 RT + TMZ: 23.3 3-week RT/TMZ + TMZ: 16.0 TMZ upfront: 8.3 RT alone: 6.0 In *MGMT*p-met vs. unmet pts: 6-week RT/TMZ + TMZ: 44.8 vs. 10.7 3-week RT/TMZ + TMZ: 27.1 vs. 12.9

Abbreviations: BSC, best supportive care; CAD, coronary artery disease; CCI, Charlson Comorbidity Index; CR, complete resection; CT, chemotherapy; DM, diabetes mellitus; DVT, deep venous thrombosis; EOR, extent of resection; pts, patients; KPS, Karnofsky Performance Status; GTR, gross-total resection; met, methylated; *MGMT*, O(6)-methylguanyl DNA methyltransferase promoter; n.a., not available (i.e., not specified in the paper); OS, overall survival; PFS, progression-free survival; pts, patients; RT, radiotherapy; STR, subtotal resection; TMZ, temozolomide; unmet, unmethylated; wks, weeks.

## Data Availability

Not applicable.

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
