# Peer review of "Elderly Gliobastoma Patients: The Impact of Surgery and Adjuvant Treatments on Survival: A Single Institution Experience"

_brainsci, 2022, doi:10.3390/brainsci12050632_

Round 1

Reviewer 1 Report

First, I had not read the manuscript, since it seemed to me that this was the case when it was enough to read the abstract. Indeed,  what did the authors find? That TMTG methylation, gross resection and better performance increase survival, and that the survival of the elderly is worse than that of the younger. But all this has long been known. This article can be summarized in one short sentence: "Our clinical experience has shown no discrepancies with existing data."

Then I did read the manuscript and found a fairly significant study with a large amount of clinical data, which reads quite interestingly, but this did not change the conclusions drawn from the abstract. Moreover, there is a lot of conflicting data in the study, especially regarding the combination of adjuvant regimens and MTMG methylation, but the authors do not bother to find explanations, limiting themselves to references to similar precedents. In addition, these explanations do not seem possible, because – as indicated in the title – this is not a structured study, but a clinical observation, i.e. a heterogeneous data set that does not allow explanations and generalizations.

Author Response

Dear Reviewer, we substantially agree with your consideration that our clinical study does not contain significant differences in comparison to previous studies. However, we tried to perform a more detailed analysis of prognostic factors as stated at the end of the Discussion.

Reviewer 2 Report

The article by Bruno et al. provides a case series of elderly patients with GBM that aims to identify patterns of care and survival at a single institution in Turin, Italy. By exploring several clinical and molecular features, the authors found a group of patients with MGMT gene promoter methylation would benefit from GTR and RT with TMZ for overall survival. This study is on a topic of relevance and general interest to the readers of the journal. The findings are interesting and the manuscript is well written. Though interpreting the data is confusing in several places throughout the paper; conclusions are, sometimes, not sufficiently supported by data. Authors should include a better description of the study background and relevance of the study. Plus, a detailed description of the analysis and statistical methods is necessary.

  1. Authors should clearly state the objective and the main findings of the study.
  2. I would like to suggest to the authors to divide the material and methods into subsections divide the material and methods into subsections and clearly detail the type of analysis performed, describe the statistical methods used, along with the comparisons performed. The ethical approval reference is missing.
  3. Result section lacks structure and flow. It would be helpful to organize results into subsections according to the analysis performed. Please, do reference all the tables during the description of the results, and not only at the end of several paragraphs.
  4. Table 1 details several information, however only patient demographics, seizure history, extent of resection, and management after surgery is described. In this case, only this information should be in this table. The remaining information should be included in new tables according to the description of the results.
    1. Why is the line 65-60 years separated from other ages? Authors should include “n, %” to the median age info.
    2. Line 86: results description does not match values from table 1: “32 patients (23.7%) older than 75 years and 9 (6.6%) older than 80 years” vs “Table 1: 75-79 years – 23; >=80 years- 6.7%” Please check.
  5. Table 2 refers to clinical complications within 1 month from surgery or 1 week before and after the surgery? Again, why is “status epilepticus” separated from the other features?
    1. Lines 102-104: please provide detailed info.
  6. Could the authors provide Table 4 with better resolution and the same consistency as the other ones? The authors only described the statistically different results without mentioning the other analysis.
  7. Regarding patient comorbidities, does any patient has a history of allergies?
  8. Lines 285-290: could the authors provide a table with this data?
  9. Could the authors provide a table with the factors adversely affecting median overall survival and overall survival?
  10. Authors identified a “subgroup of patients with favorable clinical and molecular features, who significantly benefit from large surgical resection and radio-chemotherapy”, nevertheless never clearly mentioned which group is this! Moreover:
    1. Is this subgroup sex-based?
    2. Did the authors take into consideration the gender of the patients for their analysis? Could this parameter alter the results shown here?
  11. Lines 291-294: what is the purpose of table 7 in this section?

Reviewer 3 Report

Minor:

  • Paragraph 91-95 “After surgery, 37 patients (27.4%) underwent hypofractioned (3 weeks) concurrent radio-chemotherapy with temozolomide (TMZ) followed by adjuvant TMZ; 33 (24.4%) underwent long-course (6 weeks) concurrent radio-chemotherapy with temozolomide followed by adjuvant TMZ; radiotherapy (RT) alone, upfront chemotherapy with TMZ, and RT followed by TMZ accounted for 14 (10.4%), 14 (10.4%), and 13 (9.6%)”: add Gy unit (radiation dose) here.
  • Paragraph 131-135 “Overall, patients with MGMTp methylation were more likely to undergo combined radiochemotherapy strategies than single RT, accounting for 19/30 (63.3%) and 19/35 (54.3%) of patients treated with standard 6-weeks and hypofractioned 3-weeks RT/TMZ + TMZ, respectively, as compared to 7/14 (50.0%) and 4/12 (33.3%) of those treated with RT alone or RT followed by TMZ.”: Correct those data here. According to your Table 4, data should be 57.6%, 51.4%, 50%, 30.8%.

Major:

  • Table 4: What is Palliation in the Table 4? Does it mean only surgery without any TMZ and RT ? Do other treatments include surgery with TMZ or RT? If it is, please write a definition for Palliation under the Table 4. And about Gros-total resection, Palliation has higher data (12.5%) than TMZ upfront (0%). It sounds not reasonable. Please check the calculation, and how do you define the data? If the calculation of TMZ upfront is different from other treatments, please write an explanation under the Table 4.
  • Paragraph 136-138 “Finally, CCI ≥ 3 prevailed among patients undergoing concomitant radiochemotherapy as compared to other strategies (45/70, 64.3% versus 34/65, 52.3%), even if not significantly”: Here the data do not match the number in the Table 4. How do you calculate these data, 64.3% and 52.3% ? According to your Table 4, other treatments caused more people with CCI ≥ 3 (prevailed; 11+5+8+7=31/65 = 47.7%) than concomitant radiochemotherapy (fewer people have CCI ≥ 3; 19+6=25/70 = 35%). I think your discussion seems contrary.
  • Paragraph 145-147 “Thirty-one out of 111 patients undergoing RT and / or TMZ reported adverse events (27.9%). Combined short or standard-course radiochemotherapy had a higher rate of adverse events (17/37, 45.9%; 8/33, 24.2%) as compared to TMZ upfront (3/14, 21.4%), RT alone (2/14, 14.3%), and RT followed by TMZ (1/13, 7.7%) (p = 0.034).”: You said combined radiochemotherapy caused a higher rate of adverse events, but your Table 4 said that patients with Combined radiochemotherapy have higher survival rates, 3-week: mPFS= 10.3, mOS=16; 6-week: mPFS= 18.8, mOS=28.4, RT+TMZ: mPFS= 12.7, mOS=23.2 than single therapies, TMZ upfront: mPFS= 3.3, mOS=8.3; RT alone: mPFS=3.9, mOS=6.0. Your biostatistics data and predictions sound not reasonable. Did you do other medical treatments to save patients’ lives, so the adverse events did not affect or threaten patients? And so the adverse events did not affect your mPFS and mOS data? I feel your sampling has issues causing biostatistics errors. Why can’t you sample the same population for each treatment?
  • Paragraph 156-158 “Forty-one patients out of 135 (30.3%) underwent a second line treatment, which consisted in: lomustine / procarbazine (10/41, 24.4%), regorafenib (10/41, 24.4%), fotemustine (9/41, 22.0%), TMZ rechallenge (6/41, 14.6%), bevacizumab (4/41, 9.8%), salvage radiotherapy (2/41, 4.9%).”: Does it mean that the mPFS and mOS were calculated after the second treamtments ? if it is, it looks like your mPFS and mOS data may be affected by the second line treatments. I feel your mPFS and mOS cannot be directly related to your proposed first line treatments (3-week TMZ/RT; 6-week TMZ/RT; RT alone; TMZ upfront; TMZ+RT). Your mPFS and mOS data may have some bias errors. And your Table 4 does not express summary results well. You need to consider the impact factor of the second line treatments and include them in the Table 4.
  • Paragraph 164-165 “Patients 65-69-year-old had a mPFS of 7.7 months and a mOS of 11.0 months, whereas those 70-74-year-old had mPFS 9.3 months and mOS 10.7 months, and those 75-79-year-old had mPFS 4.2 months and mOS 8.8 months (not significant).”: Where is the resulting table for these data ? Did they accept the same treatments or different ? Here you did not describe.
  • Paragraph 170-170 “MGMTp methylation was associated 170 with significantly better mPFS (11.7 versus 6.0 months, p < 0.001) and mOS (19.0 versus 171 8 months, p < 0.001).”: According to your Table 4, your data of MGMTp methylation do not show accordant and associated with significantly better mPFS and mOS. For example, in order, upfront TMZ (78.6%)-mOS(8.3); 6-weeks RT/TMZ+TMZ(57.6%)-mOS(28.4); 3-week RT/TMZ+TMZ (51.4%)-mOS(16); RT alone (50%)-mOS(6); Palliation (29.2%)-mOS(3.4); RT+TMZ (30.8%)-mOS(23.2). I can only said that patients who accepted RT and TMZ (no single TMZ, RT and Palliation) can survive.
  • Paragraph 172-174 “.EOR displayed a significant impact on outcome, being mPFS and 172 mOS longer for patients undergoing gross-total resection (18.8 and 32.8 months, respectively) than subtotal resection (6.5 and 9.9 months) or biopsy (1.0 and 6.0 months) (p = 174 0.002 for mPFS and p < 0.001 for mOS models, respectively)”: Where is their resulting table ? You have to indicate their corresponding data table here.
  • Paragraph 186-196 “The presence of MGMTp methylation correlated with a significantly longer mPFS and mOS among patients undergoing 6-weeks RT/TMZ + TMZ (mPFS: 20.8 versus 9.6 months, p < 0.001; mOS: 44.8 versus 10.7 months, p < 0.001) and 3-weeks RT/TMZ (mPFS: 19.5 versus 7.6 months, p < 0.001; 189 mOS: 27.1 versus 12.9 months, p < 0.002); also, a trend for better mOS was seen among 190 patients with MGMTp methylation treated with upfront TMZ (8.3 versus 4.6 months, p = 0.090); conversely, among patients undergoing RT alone or RT followed by TMZ the pres-192 ence of MGMTp methylation did not significantly affect the outcome. In particular, 193 MGMTp-unmethylated patients had a better survival when treated with hypofractioned RT as compared to other modalities (mOS: 12.9, 10.7, 6.0, and 4.6 months in the 3-weeks RT/TMZ + TMZ, 6-weeks RT/TMZ + TMZ, hypofractioned RT alone, and upfront TMZ 196 groups, respectively).”: Here, please indicate which resulting table corresponds to the description. If there is no table, you have to make one table for these descriptions.
  • About the section of 4. Discussion, these authors described and discussed details and interpretation, but these author did not provide a clear summary table which can express each stage of age corresponding and associated their CCI, MGMTp methylation, mPFS and mOS, etc. If authors have such tables, they have to write the indication (refer to which tables) for reviewers to easily review. Many data in the description do not directly match the data in the tables. It is hard to understand.
  • There are many other impact factors to affect mPFS and mOS, but these authors cannot express well in resulting table. The Table 4 cannot express a reasonable relationship between indexes and mPFS and mOS. Authors need to provide more clear summary tables. And sampling may have some issues. Some of the index results do not match predictions and mPFS and mOS.

Round 2

Reviewer 2 Report

The authors have addressed most of my comments, and I would like to congratulate them on this revised form of the manuscript. Now acceptable for publication.

However, the following points need to be addressed prior to publication:

Please (and again), correlate the description of your results with a specific Table(s) (both in the Results and the Discussion sections). It's very unsettling to read a paragraph of 5-10 sentences with no reference to a specific "Result", especially in such descriptive work! E.g.:

  • Results from sections 3.6; 3.7.1; and 3.7.2: please add the according Table(s)? Is Section 3.7.3 all detailed in Table 6?
  • Discussion—please add the number of the table(s) that are being discussed: Line 292; Line 295; Line 305; Line 314; Line 334; Line 340; Line 34; Line 353; Line 357.

Please, carefully proofread the manuscript to minimize typos, and grammatical errors, maintain writing consistency, and avoid redundancy. E.g.:

  • “next future”
  • starting a sentence as “135” vs “one hundred thirty-five” or with or without abbreviations
  • “radio-chemotherapy” vs “radiochemotherapy”
  • “TMZ” vs “temozolomide”

Congratulations.

Author Response

Dear Reviewer, 

thank you so much for your useful considerations and support. 

You will find our answers to your queries attached herewith.

Best regards, 

The corresponding Author.
